# Prognostic Value of Metabolic, Volumetric and Textural Parameters of Baseline [^18^F]FDG PET/CT in Early Triple-Negative Breast Cancer

**DOI:** 10.3390/cancers14030637

**Published:** 2022-01-27

**Authors:** Clément Bouron, Clara Mathie, Valérie Seegers, Olivier Morel, Pascal Jézéquel, Hamza Lasla, Camille Guillerminet, Sylvie Girault, Marie Lacombe, Avigaelle Sher, Franck Lacoeuille, Anne Patsouris, Aude Testard

**Affiliations:** 1Department of Nuclear Medicine, ICO Pays de la Loire, 15 rue André Boquel, 49055 Angers, France; olivier.morel@ico.unicancer.fr (O.M.); camille.guillerminet@ico.unicancer.fr (C.G.); sylvie.girault@ico.unicancer.fr (S.G.); marie.lacombe@ico.unicancer.fr (M.L.); avigaelle.sher@ico.unicancer.fr (A.S.); aude.testard@ico.unicancer.fr (A.T.); 2Department of Nuclear Medicine, University Hospital of Angers, 4 rue Larrey, 49100 Angers, France; FrLacoeuille@chu-angers.fr; 3Department of Medical Oncology, ICO Pays de la Loire, 15 rue André Boquel, 49055 Angers, France; clara.mathie04@gmail.com (C.M.); anne.patsouris@ico.unicancer.fr (A.P.); 4Research and Statistics Department, ICO Pays de la Loire, 15 rue André Boquel, 49055 Angers, France; valerie.seegers@ico.unicancer.fr; 5Omics Data Science Unit, ICO Pays de la Loire, Bd Jacques Monod, CEDEX, 44805 Saint-Herblain, France; pascal.jezequel@ico.unicancer.fr (P.J.); hamza.lasla@ico.unicancer.fr (H.L.); 6CRCINA, UMR 1232 INSERM, Université de Nantes, Université d’Angers, Institut de Recherche en Santé, 8 Quai Moncousu—BP 70721, CEDEX 1, 44007 Nantes, France; 7Department of Medical Physics, ICO Pays de la Loire, 15 rue André Boquel, 49055 Angers, France; 8CRCINA, University of Nantes and Angers, INSERM UMR1232 équipe 17, 49055 Angers, France; 9INSERM UMR1232 équipe 12, 49055 Angers, France

**Keywords:** TNBC, [^18^F]FDG PET/CT, prognosis, textural features

## Abstract

**Simple Summary:**

The aim of this study was to evaluate PET/CT parameters to determine different prognostic groups in TNBC, in order to select patients with a high risk of relapse, for whom therapeutic escalation can be considered. We have demonstrated that the MTV, TLG and entropy of the primary breast lesion could be of interest to predict the prognostic outcome of TNBC patients.

**Abstract:**

(1) Background: triple-negative breast cancer (TNBC) remains a clinical and therapeutic challenge primarily affecting young women with poor prognosis. TNBC is currently treated as a single entity but presents a very diverse profile in terms of prognosis and response to treatment. Positron emission tomography/computed tomography (PET/CT) with ^18^F-fluorodeoxyglucose ([^18^F]FDG) is gaining importance for the staging of breast cancers. TNBCs often show high [^18^F]FDG uptake and some studies have suggested a prognostic value for metabolic and volumetric parameters, but no study to our knowledge has examined textural features in TNBC. The objective of this study was to evaluate the association between metabolic, volumetric and textural parameters measured at the initial [^18^F]FDG PET/CT and disease-free survival (DFS) and overall survival (OS) in patients with nonmetastatic TBNC. (2) Methods: all consecutive nonmetastatic TNBC patients who underwent a [^18^F]FDG PET/CT examination upon diagnosis between 2012 and 2018 were retrospectively included. The metabolic and volumetric parameters (SUV_max_, SUV_mean_, SUV_peak_, MTV, and TLG) and the textural features (entropy, homogeneity, SRE, LRE, LGZE, and HGZE) of the primary tumor were collected. (3) Results: 111 patients were enrolled (median follow-up: 53.6 months). In the univariate analysis, high TLG, MTV and entropy values of the primary tumor were associated with lower DFS (*p* = 0.008, *p* = 0.006 and *p* = 0.025, respectively) and lower OS (*p* = 0.002, *p* = 0.001 and *p* = 0.046, respectively). The discriminating thresholds for two-year DFS were calculated as 7.5 for MTV, 55.8 for TLG and 2.6 for entropy. The discriminating thresholds for two-year OS were calculated as 9.3 for MTV, 57.4 for TLG and 2.67 for entropy. In the multivariate analysis, lymph node involvement in PET/CT was associated with lower DFS (*p* = 0.036), and the high MTV of the primary tumor was correlated with lower OS (*p* = 0.014). (4) Conclusions: textural features associated with metabolic and volumetric parameters of baseline [^18^F]FDG PET/CT have a prognostic value for identifying high-relapse-risk groups in early TNBC patients.

## 1. Introduction

Breast cancer is the leading cause of cancer death in women worldwide [1]. Triple-negative breast cancer (TNBC) accounts for 15% of all breast carcinomas and is defined by the lack of estrogen and progesterone receptor (ER, PR) expression without HER2 amplification.

This phenotype occurs frequently in younger women (<50 years), is more prevalent in African American women [2] and is generally associated with a large tumor size, a high grade, a high mitotic index and lymph node involvement at diagnosis. TNBC presents a high risk of both local and distant recurrence, including T1 tumors [3]. The lack of a therapeutic target makes its therapeutic optimization difficult [4]. Targeted therapies provide contrasting results, albeit with the promising arrival of immunotherapy, PARP inhibitors and, more recently, antibody–drug conjugates such as sacituzumab–govitecan. Often, however, conventional cytotoxic drugs (including the use of platinum salts) remain the mainstay of current systemic management. Moreover, this phenotype presents a paradox: a high rate of response to chemotherapy in neoadjuvant settings, neighboring 50% [5], but also a high rate of distant and early recurrence. Indeed, relapses occur mainly in the first three years after initial treatment with metastatic cerebral and visceral tropism [4,6,7,8].

Although TNBCs are treated as a single entity in clinical practice, it is now accepted that they encompass significant heterogeneity in their clinical and phenotypic presentation, prognosis and response to treatment. From a molecular point of view, TNBCs share similarities with the basal-like breast cancer defined by Perou and Sorlie, BRCA1-related disease and claudin-low breast cancer [9]. Lehmann’s team described six molecular subtypes: basal-like 1 (BL1), basal-like 2 (BL2), immunomodulatory (IM), mesenchymal-like (M), mesenchymal stem-like (MSL) and luminal androgen receptor (LAR) [10], which were then refined into four tumor-specific subtypes (BL1, BL2, M and LAR) [11].

Jézéquel et al. have identified three clusters: C1 (22.4%) enriched in luminal subtypes and positive LAR; C2 (44.9%), almost a pure basal-like cluster; and C3 (32.7%), enriched in basal-like subtypes but to a lesser extent and including 26% claudin-low subtypes, with distinct prognoses [12,13,14].

In early settings, clinical and pathological prognostic factors include tumor size, lymph node involvement, high Ki67 and the presence of lymphovascular emboli [3,15,16]. Moreover, many studies demonstrate the strong prognostic role of stromal tumor-infiltrating lymphocytes (TILs) in disease-free survival (DFS) and overall survival (OS) in TNBC [17,18]. Finally, pathological complete response (pCR) is a surrogate endpoint for predicting DFS and OS, especially for TNBC [5,19]. Adjuvant capecitabine administration has shown a reduced risk of recurrence in TNBCs without pCR [20].

Positron emission tomography/computed tomography (PET/CT) with ^18^F fluorodeoxyglucose ([^18^F]FDG) is gaining importance for the staging of patients with large or locally advanced breast cancer [21]. TNBCs often show high [^18^F]FDG uptake [22,23,24,25,26,27,28], and several studies have demonstrated correlations between standardized uptake values (SUVs) and histoprognostic factors such as tumor size, grade or Ki67 [24,25,26,27,28,29,30,31,32,33,34,35]. Some studies have analyzed PET parameters’ prognostic impact in nonmetastatic breast cancers, with two meta-analyses [36,37] suggesting that Metabolic Tumor Volume (MTV) and Total Lesion Glycolysis (TLG) are correlated with DFS and OS. In TNBC, few studies have focused on these parameters. Ohara et al. [22] found that the maximum SUV (SUV_max_) of the primary breast lesion was correlated with DFS. Kim et al. [38] demonstrated that the metabolic and volumetric parameters of lymph node involvement (SUV_max_, MTV and TLG) had a prognostic impact, whereas no significant result was shown for these parameters in the primary breast tumor.

Textural analysis, also known as “radiomics”, is opening up new perspectives on tumor imaging analysis [39,40], such as decoding the tumor phenotype on initial examination, evaluating the pathological response after neoadjuvant therapy or eventually predicting the outcomes of the diseases. Of the many features developed in the literature, Orlhac et al. [41] identified six textural parameters which seem to be reproducible, robust and independent. Soussan et al. [42] found that TNBCs were associated with High Gray-level Run Emphasis (HGRE) and with low homogeneity. In a study by Groheux et al. [43] on ER+/HER2- breast cancers, MTV had a prognostic value, whereas textural parameters such as entropy and homogeneity were not significant. Other studies have looked at breast cancers with discordant results, often related to methodological differences due to a lack of standardized practices [44,45,46,47,48,49]. Furthermore, textural parameters have prompted interest, with potentially promising prognostic values as suggested in other cancers [50] such as head-and-neck [51,52,53,54] and lung carcinomas [55,56], but no study to our knowledge has evaluated these parameters’ prognostic value in TNBC.

The main purpose of this study was to identify prognostic factors using the metabolic, volumetric and textural parameters upon [^18^F]FDG PET/CT in early TNBC patients with DFS.

The secondary objectives were to correlate the metabolic, volumetric and textural features of [^18^F]FDG with OS and clinical and pathological factors.

## 2. Materials and Methods

### 2.1. Study Design

This retrospective single-center study included all consecutive nonmetastatic TNBC patients who underwent [^18^F]FDG PET/CT at diagnosis between January 2012 and December 2018 at the Institut de Cancérologie de l’Ouest (ICO), Angers (France).

Inclusion criteria were: unilateral TNBC confirmed by immunohistochemistry (IHC) (defined by ER and PR < 10%, HER2 0 or 1+ in IHC or HER2++ in IHC with negative in situ hybridization [ISH]); and tumor classified as cT1 to T4, cN0 or cN+ using the TNM classification. Patients had to be aged over 18.

Exclusion criteria were: HER2 tumors positive on ISH; patients who underwent surgery on the primitive tumor before PET/CT; patients who had distant metastasis; primary breast tumor with low [^18^F]FDG uptake (SUV_max_ < 2.5) (these lesions were not discernable from metabolic background of the breast tissue); bilateral breast tumor; previous history of breast cancer (except contralateral in situ breast carcinoma); history of cancer <5 years (except basal-cell carcinoma); current cancer treated with chemotherapy; immunotherapy or radiotherapy; and opposition to data analysis.

Every patient received an information sheet. When authorized, each medical record was analyzed to extract the necessary data. We acquired information on the death of the patients in the medical files with the death certificate. Most of the time, the patients had a follow-up with the Institute’s oncologists every 6 months for 5 years after treatment. The study protocol was approved by an independent ethics committee (registered under number 2019/115).

### 2.2. Clinical and Histological Parameters

We retrieved patient characteristics (age, history of cancer), tumor parameters (clinical size confirmed by ultrasound, inflammatory nature and clinical axillary lymph node involvement) and histological parameters (histological type, mSBR grade, presence of in situ carcinoma, Ki67, unifocal or multifocal tumor, CK5/6, presence of lymphovascular emboli and presence of significant tumor-infiltrating lymphocytes (>30%)). Data on therapeutic modalities (type of chemotherapy, type of breast and axillary surgery and type of radiotherapy) were collected.

### 2.3. [^18^F]FDG PET/CT Examination

Patients fasted for 6 h and their blood glucose level had to be less than 7.0 mmol/L. The [^18^F]FDG (2–3 MBq/kg) was administered intravenously. Acquisition started 60 min (±5 min) after injection and was performed from the top of the skull to the mid-thigh level, arm raised. A low-dose CT (slice thickness: 3.75 mm) was taken according to a standardized protocol. Examinations were performed using three types of PET/CT systems (GE Healthcare USA, Inc., Chicago, IL, USA): Discovery ST (DST), Discovery 690 (D690) and Discovery IQ 5 rings (DIQ5), ensuring the standardization and harmonization of imaging procedures. Time for bed scans were, respectively, 2 mn 30 (DST and D690) and 2 mn (DIQ5). To allow semiquantitative data pooling, we evaluated the contrast recovery coefficients of each system’s PET reconstructed images with an EANM/EARL-like method.

For image quality control measurements, the NEMA NU2-2012 image quality phantom was required, with six fillable spheres having internal diameters of 10 mm, 13 mm, 17 mm, 22 mm, 28 mm and 37 mm, positioned coaxially around a lung insert. The phantom background compartment and spherical inserts were filled with [^18^F]FDG solution, resulting in a sphere-to-background ratio of 5:1.

A visual analysis did not show any difference (Figure 1), which was confirmed by analysis of the recovery contrast curves (Figure 2).

### 2.4. [^18^F]FDG PET/CT Imaging Analysis

Using DOSIsoft software (Planet Onco v2.0), under the supervision of an experienced nuclear medicine physician, a single junior nuclear medicine physician drew for each patient a large three-dimensional region of interest (3D-ROI) around the primitive breast lesion. Lymph node involvement in PET/CT (N+_PET_) was considered to be any well-defined focus with [^18^F]FDG uptake clearly higher than the surrounding background, independently of the histological analysis.

According to two previous studies evaluating TNBC [38,57], a segmentation method using a threshold of 2.5 of the SUV (SUV2.5) was used to enroll all voxels of interest and delimitate the volume of interest (VOI).

Metabolic and volumetric parameters (SUV, MTV and TLG) were analyzed for the primary breast lesion. They are described here:-SUV: calculated using the following equation:
SUV = (tissue radioactivity [Bq]/tissue weight [g])/(injected activity [Bq]/body weight [g]).
○Max: highest value of a voxel in the VOI○Mean: mean value of all voxels in the VOI○Peak: mean value of all voxels in 1 cm^3^ centered around the highest voxel value-MTV: considered as the volume of all voxels included in the VOI-TLG: defined by the formula SUV_mean_ × MTV

Textural features were calculated on the delineated tumor volume of the primary breast lesion, extracted from the matrix using absolute discretization between 0 and 20 SUV units with 64 gray levels. Six textural features were selected and analyzed according to Orlhac’s work (41):-Homogeneity: measures the local homogeneity of a pixel pair—homogeneity is expected to be large if each pixel pair’s gray levels are similar.-Entropy: measures the randomness of a gray-level distribution—the entropy is expected to be high if the gray levels are distributed randomly throughout the tumor region.-Short-Run Emphasis (SRE): measures the distribution of short series—the value is expected to be large if the number of short series is high.-Long-Run Emphasis (LRE): measures the distribution of long series—the value is expected to be large if the number of long series is high.-Low-Gray-level Zone Emphasis (LGZE): measures the distribution of low-gray-level zones.-High-Gray-level Zone Emphasis (HGZE): measures the distribution of high-gray-level zones.

In cases of multifocal breast tumor, textural features were extracted from the most [^18^F]FDG-avid lesion (highest SUV_max_). PET parameters for multifocal breast tumors were the sum of each lesion (MTV) or the sum of weighted averages of each lesion (SUV_mean_ and TLG).

### 2.5. Statistical Analysis

To compare the raw values of each radiomics feature between the three types of PET/CT systems, Kruskal–Wallis nonparametric tests were performed.

Principal component analysis (PCA) was used to explore technical sources of variation (i.e., batch effect) because radiomics features were notably measured using three types of PET/CT systems. With this aim, data were evaluated using different harmonization methods. They included: ComBat (https://forlhac.shinyapps.io/Shiny_ComBat/, accessed on 5 May 2021), remove batch effect (RBE) and variable standardization [58,59,60]. Guided PCA (gPCA) was used to evaluate the three harmonization techniques [61].

Quantitative variables were summarized using mean, standard deviation (SD), median, minimum, maximum and interquartile range (IQR) and were compared using Student’s t-test or Mann–Whitney, as appropriate. To ensure consistency in applying inferential analyses, the PET/CT values with non-Gaussian distribution were log transformed. Binary and categorical variables were summarized using count and percentage and were compared using Chi-square test or Fisher’s exact test if appropriate.

OS was defined between diagnosis and death or censored date for living patients. DFS was defined between diagnosis and relapse (RECIST criteria) or censored date for living patients without relapse. Estimates and survival curves were computed using the Kaplan–Meier method.

To identify the factors associated with DFS and OS, univariate Cox regression models were used. Variables significantly associated with endpoints were included in a complete multivariate model. A stepwise backward selection model based on AIC was used to define the final multivariate model and estimate the Hazard Ratio (HR) and 95% confidence interval (95% CI). No adjustment of the *p*-value for multiplicity of tests was realized in this exploratory retrospective study.

To formalize the association between PET/CT parameters and DFS and OS in clinical practice, thresholds were identified using Receiver Operating Characteristic (ROC) curves for DFS and OS at 24 months. Youden’s index was used to identify the better threshold from sensitivity and specificity results for each parameter separately. Survival curves were computed using the Kaplan–Meier method, and a log-rank test was performed to compare DFS and OS between groups.

We considered a two-sided *p*-value of less than 0.05 to be statistically significant.

Analyses were conducted using R freeware version 3.6.2.

(https://www.R-project.org/, accessed on 1 July 2020) and FactoMineR package version 2.2 (http://www.jstatsoft.org/v25/i01/, accessed on 1 July 2020).

## 3. Results

### 3.1. Patient Inclusion

Between January 2012 and December 2018, 679 breast cancer patients had an initial PET/CT examination at the ICO center (Figure 3). Of these, 568 patients were excluded (245 had a hormone-receptor-positive and/or a HER2-positive tumor, 277 had a previous history of cancer, 30 patients had metastatic cancer at diagnosis, 13 had had breast surgery before the PET/CT examination and 3 patients did not present hypermetabolic lesions in the PET/CT). Finally, 111 TNBC patients were enrolled in this study: 45 with Discovery ST, 49 with Discovery 690 and 17 with Discovery IQ 5 rings.

### 3.2. Comparison of Raw and Harmonized Data

There was no significant difference in metabolic, volumetric and textural feature values between the three types of PET/CT scanners (*p* > 0.05) (Figure 4). For the PCA of the nonharmonized data from the three kinds of systems, only one homogeneous group was displayed (Figure 5a). This descriptive result was confirmed by gPCA (δ = 0; *p* = 1), which signified that there was no batch effect due to the type of PET/CT system. For the three harmonization methods, the projection of the three types of PET/CT scanners in the first PCA plane also showed a homogeneous group (Figure 5b–d). These latter results demonstrated that there was no need to harmonize the data obtained from the different systems.

### 3.3. Patient Characteristics and PET/CT Baseline Parameters

The median age was 52.6 years. Primary breast lesions were mainly T1–T2 stage (71.8%), and 50 patients (45%) had positive lymph node (LN) involvement in the PET/CT (N+_PET_). Most of the tumors were grade 3 (92.8%), with a Ki67 ≥ 20% (91.3%). Of the patients, 72.9% received neoadjuvant chemotherapy, 65.8% conservative surgery, 59.5% axillary dissection and 94.5% radiotherapy. Patient characteristics are summarized in Table 1.

The metabolic and volumetric parameters of the primary breast tumors were mostly high, with the mean SUV_max_ measured as 14.6, the mean MTV as 22.6 cm^3^ and the mean TLG as 191.3. These parameters are summarized in Table 2.

The median follow-up was 53.6 months (95% CI: 40.7–61.6). The median DFS was not reached (Figure 6). Among the 111 patients, 20 died (18%) and 34 relapsed (30.6%). Of these 34 patients, 23 presented with metastatic relapse (67.6%), 9 with locoregional relapse (26.5%), 1 with contralateral relapse (2.9%) and 1 with a second cancer (cervical) (2.9%). Most relapses were observed in the first two years after treatment (30/34 patients). Of the deaths, 22 out of 30 occurred in the first three years (Figure 7).

### 3.4. PET/CT Features with Clinical and Histological Factors

Homogeneity was correlated with a low grade (*p* = 0.004) and with no PET/CT lymph node involvement (N0_PET_) (*p* = 0.037). MTV was associated with large tumors (*p* < 0.001), inflammatory carcinoma (*p* < 0.001) and with clinical and PET/CT lymph node involvement (*p* < 0.001). Entropy was associated with large tumors (*p* < 0.001), a higher grade (*p* = 0.02), clinical and PET/CT lymph node involvement (*p* = 0.027 and *p* = 0.001, respectively) and inflammatory tumors (*p* < 0.001). SRE was only associated with a higher grade (*p* = 0.044) (Table 3).

### 3.5. Tumor Characteristics, Therapeutic Management and PET/CT Features with Prognostic Factors

Regarding the clinical and histological factors, in the univariate analysis, the size of the primary tumor and multifocality were only associated with OS (*p* = 0.022 and *p* < 0.001, respectively). Inflammatory carcinoma and clinical lymph node involvement were associated with DFS (*p* = 0.025 and *p* = 0.001, respectively) and OS (*p* = 0.004 and *p* = 0.038, respectively) (Table 4).

Regarding PET/CT parameters in univariate analysis, high TLG, MTV and entropy values of the primary tumor were poor prognostic factors for DFS (*p* = 0.008, *p* = 0.006 and *p* = 0.025, respectively) and OS (*p* = 0.002, *p* = 0.001 and *p* = 0.046, respectively). Hypermetabolic lymph nodes upon PET/CT (N+_PET_) were associated with lower DFS (*p* = 0.004) and lower OS (*p* = 0.009) (Table 5).

In the multivariate analysis, adjusting for clinical, histological and PET features, N+_PET_ was correlated with lower DFS (*p* = 0.036), and high MTV was correlated with lower OS (*p* = 0.014) (Table 6).

The discriminating thresholds for two-year DFS were calculated as 7.5 for MTV, 55.8 for TLG and 2.6 for entropy (Figure 7, Figure 8 and Figure 9), with a sensitivity of 70.8%, 63% and 78.1%, respectively, and a specificity of 65.1%, 68.3% and 60.3%, respectively. The discriminating thresholds for two-year OS were calculated as 9.3 for MTV, 57.4 for TLG and 2.67 for entropy (Figure 8, Figure 9 and Figure 10), with a sensitivity of 92.4%, 92.4% and 83.1%, respectively, and a specificity of 66.7%, 68% and 64%, respectively. Using these thresholds, we found a significant difference in DFS at 24 months for the three parameters and a difference in OS for MTV and TLG. However, the difference was not significant for OS at 24 months, according to the 2.67 threshold for entropy.

## 4. Discussion

To our knowledge, this study is the first to evaluate the prognostic impact of textural features in TNBC. All consecutive nonmetastatic primary TNBC patients undergoing a PET/CT examination before any treatment were carefully selected. All primary breast lesions were segmented by the same method using a threshold of SUV2.5. In our study, tumor characteristics (grade, size and lymph node involvement) and outcome data were representative of the TNBC population with a high rate of metastatic relapse occurring in the first two years [4,6,7].

We demonstrated that the volumetric parameters of the primary breast cancer (MTV and TLG) were predictive of recurrence and death. These results are consistent with early studies, which found that volumetric parameters had a significant impact on prognosis in breast cancers [62,63], such as a recent meta-analysis of 975 primary tumors by Pak et al. [36] demonstrating that a high MTV or TLG leads to a higher risk of adverse events. Conversely, Kim et al. did not find any correlation between the metabolic parameters of the primary breast tumor and prognostic factors in 222 TNBC patients [38].

Other metabolic parameters studied in the primary breast tumors, such as the SUV_max_, were not correlated with DFS and OS in our study. Indeed, SUV_max_ is a controversial prognostic factor. In TNBC, two studies have demonstrated that the primary breast tumor’s SUV_max_ was correlated with DFS [57] and had a poor prognostic value if [^18^F]FDG uptake was over 8.6 [22], whereas our results echoed Kim et al. [38], who did not find a prognostic impact for this parameter.

SUV is considered as a variable parameter whose result is the product of the metabolic activity normalized to the patient’s weight and blood glucose level. It is influenced by many factors (type of machine, acquisition and reconstruction protocol, lesion size, patient movements, etc.) but remains the reference for quantitatively characterizing the result of a PET/CT examination [64]. SUV_max_ reflects the value of a single pixel, which is not representative of the entire lesion set. MTV appears to be more representative of lesion impairment, but its value depends mainly on the segmentation methods [65]. TLG appears to be an interesting endpoint, including volumetric criteria weighted by the lesion’s mean metabolism.

Textural features are beginning to be studied more and more because of their potential prognostic contribution in addition to known metabolic and volumetric parameters. By evaluating the distribution, position and repetition of voxels or a group of voxels, depending on their intensity, PET/CT imaging seems to be able to characterize lesions in addition to their metabolism. Radiomics is used to decode the tumor phenotype with noninvasive imaging procedures: without biopsy, textural features could potentially be used to analyze the entire tumor lesion and to predict the response to treatment and patient outcome. It has been demonstrated that heterogeneity derived in vivo from PET images accurately reflects the heterogeneity of tracer uptake derived ex vivo from autoradiographic images [66]. In breast cancers, Umutlu et al. [67] and Krajnc et al. [68] have found that textural features have an interest for decoding tumor phenotypes. Recent studies in breast cancers have shown the potential of textural features for predicting the pathological response after neoadjuvant treatment [69,70,71]. Several studies have analyzed the correlation between textural features, pathological parameters, prognostic impact and response to therapy in multiple cancers. For example, in non-small-cell lung (NSCLC) and esophageal cancers, two parameters (intensity variability and size-zone variability), reflecting regional heterogeneity, were correlated with prognostic-free survival (PFS) [72,73]. Another study in 233 NSCLC patients demonstrated that entropy, homogeneity, dissimilarity and zone percentage were significantly predictive of PFS in the univariate model [74]. In cervical cancers, Chen et al. [75] found that HGRE had a prognostic value regarding DFS and OS. Correlation, a textural feature reflecting tumoral heterogeneity, was able to predict the pathological response and was correlated with DFS and OS in colorectal cancers [76].

Finally, multiple textural features have been developed and analyzed in the literature [50]. It has been demonstrated in some studies that TNBC correlates with heterogeneous textural features compared to other breast cancer subtypes. This can be explained by the fact that TNBCs are often aggressive tumors whose pejorative histoprognostic factors (grade, proliferation rate, size, inflammatory tumor, lymph node involvement, etc.) are correlated with some metabolic parameters and pejorative features [24,25,26,27,28,29,30,31,32,33,34,35], whereas less aggressive tumors such as ER+ breast cancers seem to be more homogeneous, as suggested by Groheux et al. [43].

As suggested by Lambin et al. [39], radiomics works could address the association between the high-throughput extraction of large amounts (>200) of quantitative features of medical images and clinical assessments. Therefore, as in genomics studies, the number of studied parameters may often exceed the number of observations (*p* >> *n*), which requires specific statistical solutions (including a strong reduction in the *p*-value threshold to be considered significant, to avoid alpha risk inflation). In this study, we focused on very fewer parameters (*p* = 11), and, since *n* > *p*, the conventional statistical methods could be applied. Rather than coming to a statistical conclusion, we aimed to explore the possible prognostic value of several metabolic parameters and textural features measured at the initial examination in a specific TNBC population.

We decided to evaluate only six textural features, according to Orlhac’s previous work [41]. They were found to be the most reproducible, robust and independent among 32 parameters. Of the six parameters evaluated in this study, only one, entropy, had a significant prognostic value regarding DFS and OS in the univariate analysis. Entropy was also correlated with pejorative histoprognostic factors such as high-grade or inflammatory tumors in our study. This textural parameter has been evaluated in other malignancies, such as head-and-neck squamous cell carcinoma [51], NSCLC [74] and melanoma [77], with a prognostic value for OS, PFS and pathological response to treatment, but no study has evaluated the prognostic impact of textural parameters in TNBC.

Textural features tend to be an interesting additional parameter of PET/CT imaging, but results are varied: studies often involve small cohorts of patients, are mostly retrospective and analyze multiple different textural parameters. There is no consensus over their use in clinical practice, so these results need to be confirmed by large prospective studies or randomized meta-analyses. It is necessary to at least develop standardized practices or guidelines for imaging analysis covering lesion segmentation methods and validated software for the extraction of textural parameters.

We have also suggested MTV and TLG thresholds for the early identification of TNBC patients with a poor prognosis in clinical practice. Jiménez-Ballvé et al. [62] proposed a MTV threshold of 19.3 cm^3^, including all breast cancer subtypes. An MTV threshold of 7.5 cm^3^ was proposed in our study, reflecting the fact that TNBC patients have a worse overall prognosis than all subtypes combined. To our knowledge, there is no study proposing thresholds for TNBC patients, except that of Ohara et al. for the SUV_max_ [22].

A limitation of this work was that it was a retrospective and monocentric study, which excluded patients with no [^18^F]FDG-avid primary breast tumor (three patients with SUV_max_ < 2.5). TNBC patients who underwent PET/CT at diagnosis mostly had proliferative biomarkers (i.e., 92.8% of patients had a grade-three tumor and 91.3% had a Ki67 ≥ 20%), meaning that even if our survival data were representative of the TNBC population, we probably did not capture all TNBCs. For example, according to Lehmann et al., Jézéquel et al. and Burstein et al., the molecular entity LAR, often characterized by low proliferation at the phenotype level, is probably underrepresented [10,12,13,14,78].

There is no consensus over segmentation methods for imaging analysis. We decided to use the threshold of 2.5 of the SUV_max_ to enroll primary lesions and lymph nodes, reflecting previous studies on TNBC [38,57], which seemed to be more representative of the metabolic volume according to eye discrimination than methods using a percentage of the SUV_max_, especially for high [^18^F]FDG-avid and heterogeneous lesions. We know that metabolic, volumetric and textural values depend on segmentation methods [79,80]. Another limitation is that we analyzed images from three different PET/CT scanners between 2012 and 2018, although the scanners were standardized in an EARL-like protocol [81]. This measurement bias was also removed by our additional study, which demonstrated that there was no variation in the metabolic, volumetric and textural features extracted from the three different PET/CT scanners.

It could be interesting to integrate clinical, multiomic and textural features in order to define distinct subgroups within triple-negative tumors and optimize their treatment. Textural analysis seems to give noninvasive information on the tumor phenotype and is potentially correlated with prognostic values in cancers. Finally, the dynamic characterization of tumor heterogeneity by assessing these parameters before and after treatment could contribute to exploring a clonal selection phenomenon with a better definition of residual treatment-resistant disease. 

## 5. Conclusions

This study confirms the prognostic impact of the metabolic parameters of primary breast tumors and the lymph node involvement in PET/CT in early TNBC patients. Entropy seems to have a potential prognostic value for identifying high-relapse-risk groups and defining patients with poor prognosis for whom therapeutic escalation can be considered, but this must be confirmed in a larger study.

## Figures and Tables

**Figure 1 cancers-14-00637-f001:**
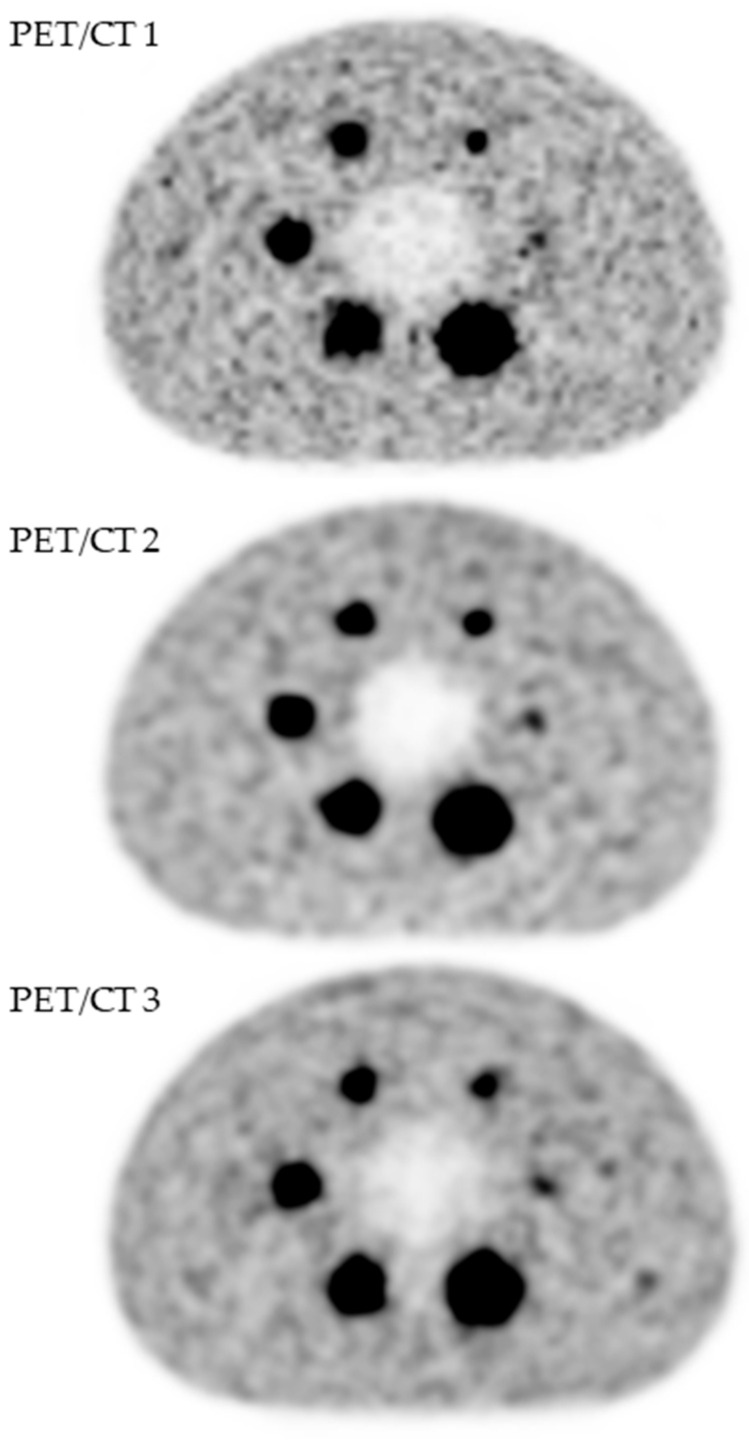
Quality assessment of [^18^F]FDG PET images obtained with three PET/CT systems (GE Healthcare USA, Inc., Chicago, IL, USA): Discovery ST (PET/CT 1), Discovery 690 (PET/CT 2) and Discovery IQ5 (PET/CT 3).

**Figure 2 cancers-14-00637-f002:**
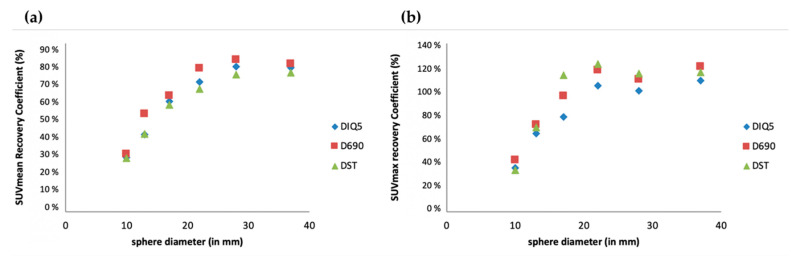
Semiquantitative assessment (SUV_mean_ (**a**) and SUV_max_ (**b**)) of [^18^F]FDG PET images obtained with three PET/CT systems (GE Healthcare USA, Inc.): Discovery ST (DST), Discovery 690 (D690) and Discovery IQ5 (DIQ5).

**Figure 3 cancers-14-00637-f003:**
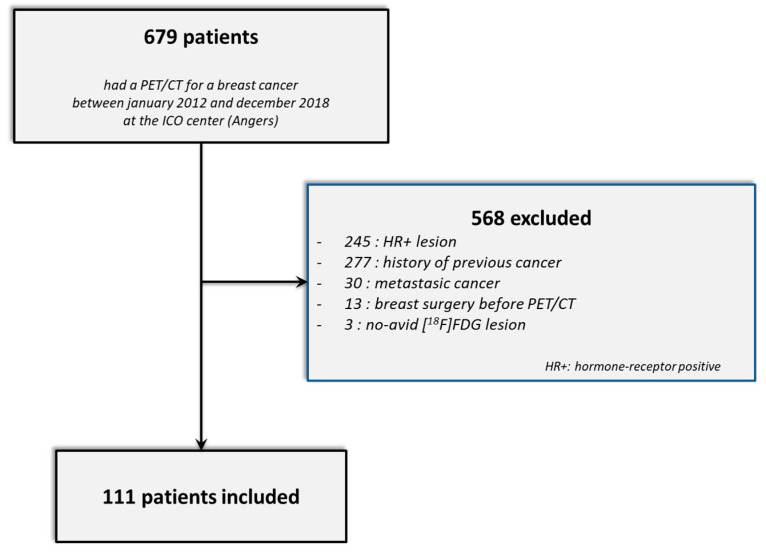
STARD flow diagram.

**Figure 4 cancers-14-00637-f004:**
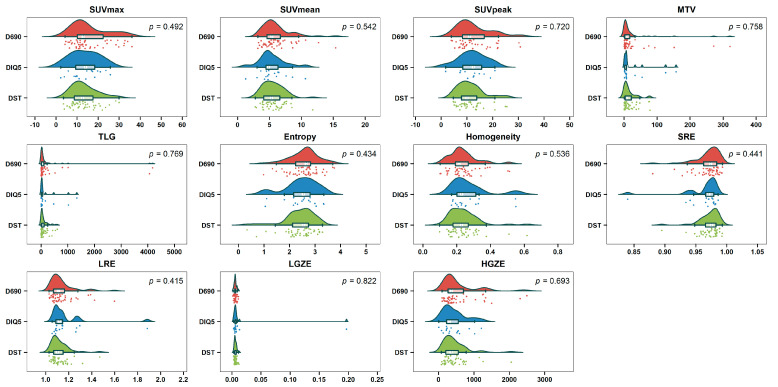
Raincloud plots of metabolic, volumetric and textural features from the three types of PET/CT system (Discovery 690 (D690); Discovery IQ 5 rings (DIQ5); Discovery ST (DST)) before harmonization.

**Figure 5 cancers-14-00637-f005:**
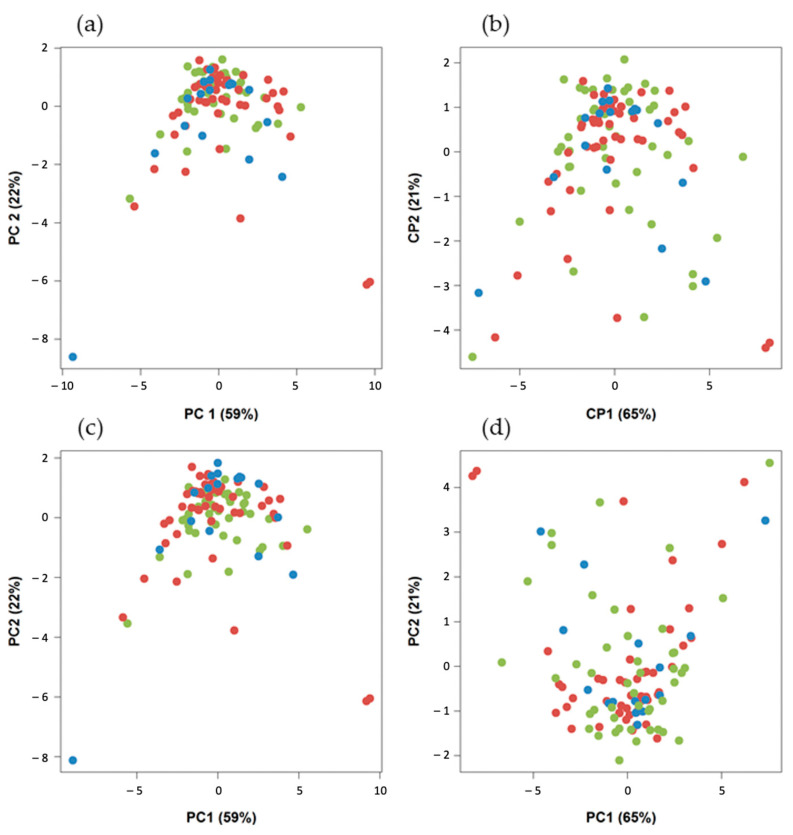
Projection of the three types of PET/CT system in the first PCA plane (Discovery 690, *n* = 49 (red); Discovery IQ 5 rings, *n* = 17 (blue); and Discovery ST, *n* = 45 (green)): (**a**) without harmonization, (**b**) with ComBat harmonization, (**c**) with batch-effect removal harmonization and (**d**) variable standardization harmonization.

**Figure 6 cancers-14-00637-f006:**
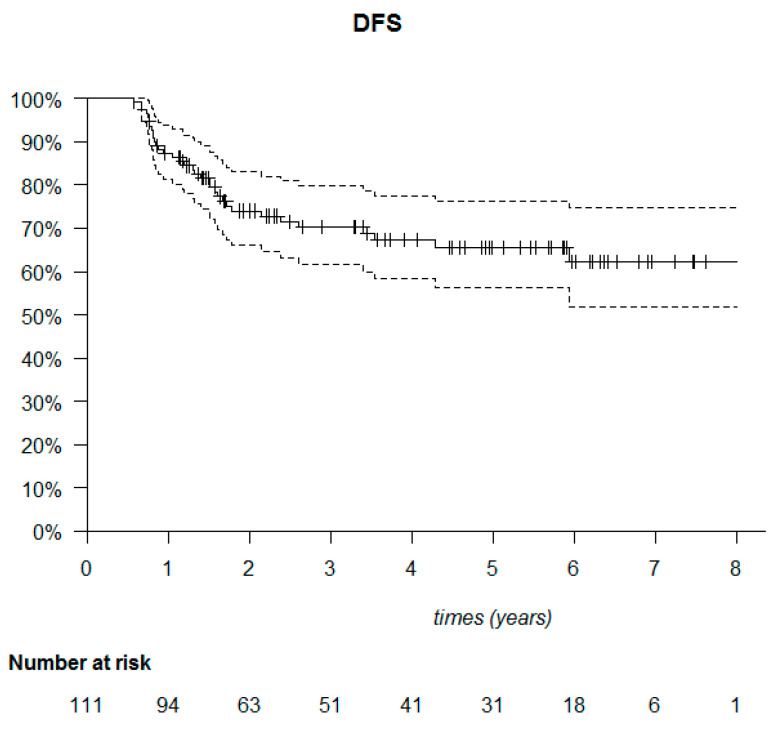
Kaplan–Meier disease-free survival (DFS) of the 111 triple-negative breast cancer patients.

**Figure 7 cancers-14-00637-f007:**
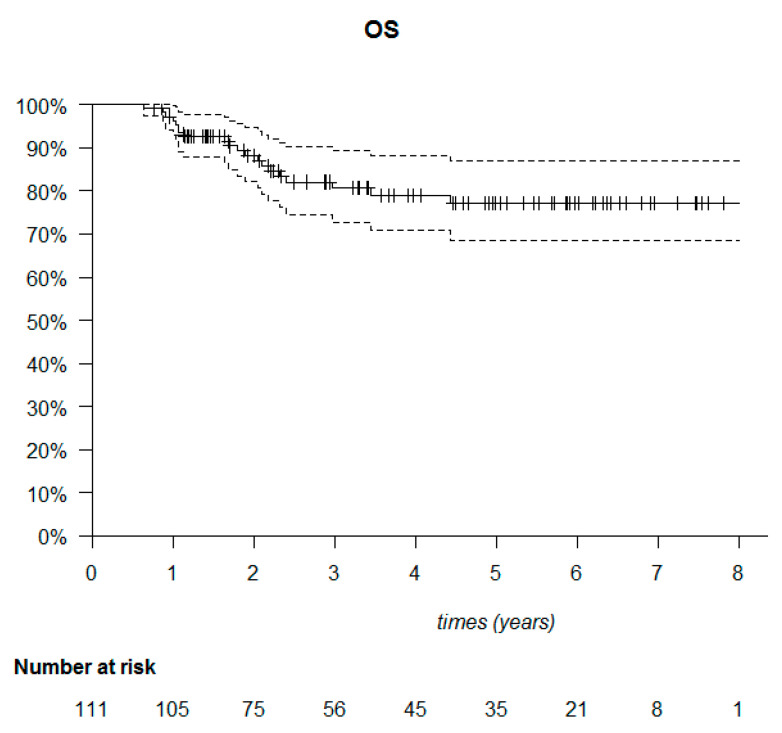
Kaplan–Meier overall survival (OS) of the 111 triple-negative breast cancer patients.

**Figure 8 cancers-14-00637-f008:**
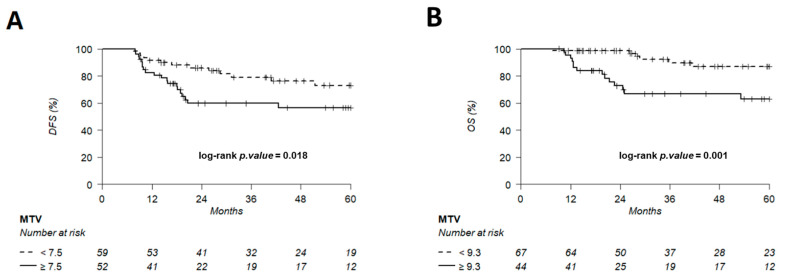
Kaplan–Meier of DFS (**A**) using an MTV threshold of 7.5 and OS (**B**) using an MTV threshold of 9.3. Thresholds are predetermined using Young’s index.

**Figure 9 cancers-14-00637-f009:**
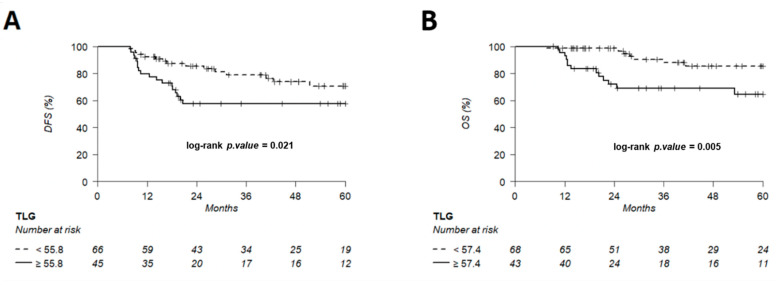
Kaplan–Meier of DFS (**A**) using a TLG threshold of 55.8 and OS (**B**) using a TLG threshold of 57.4, predetermined using Young’s index.

**Figure 10 cancers-14-00637-f010:**
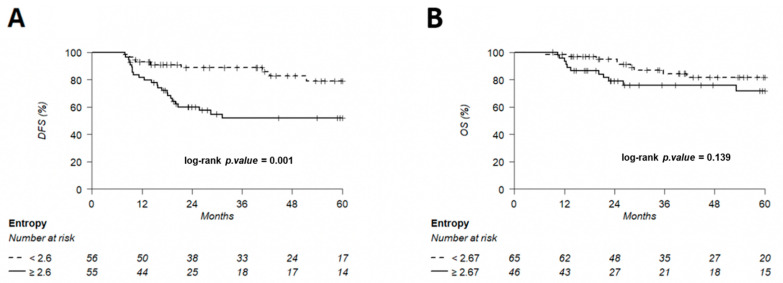
Kaplan–Meier of DFS (**A**) using an entropy threshold of 2.6 and OS (**B**) using an entropy threshold of 2.67, predetermined using Young’s index.

**Table 1 cancers-14-00637-t001:** Patient characteristics.

Patient Information	Mean [min–max]	SD
Age	52.6 [25–90]	14.7
**Tumor characteristics**	***n* = 111**	**%**
Inflammatory tumor	12	10.9
T stage		
T1–T2	80	71.8
T3–T4	31	28.2
N stage		
cN0	70	62.7
cN+	41	37.3
Ductal carcinoma	104	93.7
mSBR grade		
2	8	7.2
3	103	92.8
Mitoses (*n* = 106)		
1	5	4.7
2	17	16.0
3	84	79.2
Associated in situ carcinoma	25	27.5
Ki67 ≥ 20%	94	91.3
Unifocal tumor Multifocal tumor	86 25	77.5 22.5
Lymphovascular emboli	22	21.6
**Therapy**	***n* = 111**	**%**
Chemotherapy (CT)		
Adjuvant	29	27.1
Neoadjuvant	78	72.9
Type of CT		
Anthracyclines + taxanes	81	76.4
Without anthracyclines	3	2.8
Platinum salts	22	20.8
Tumor surgery		
Conservative	73	65.8
Radical	38	34.2
Lymph node surgery		
Dissection	66	59.5
Sentinel	45	40.5
Radiotherapy		
Yes	104	94.5
No	7	5.5
(pCR) (*n* = 74)		
Yes	22	29.7
No	52	70.3
**PET/CT imaging**	***n* = 111**	**%**
N_PET_ stage		
N0_PET_	61	55.0
N+_PET_	50	45.0

mSBR: modified Scarff–Bloom–Richardson; pCR: pathological complete response.

**Table 2 cancers-14-00637-t002:** Mean and median values of metabolic and volumetric parameters of the primary breast tumors of the 111 TNBC patients upon PET/CT.

PET Parameters	Mean (SD)	Median (IQR)
SUV_max_	14.6 (7.6)	12.8 (9.6–18.3)
SUV_mean_	5.8 (2.1)	5.3 (4.5–6.65)
SUV_peak_	12.1 (5.9)	10.9 (7.9–14.2)
MTV	22.57 (46.99)	6.8 (2.9–18.1)
TLG	191.33 (578.25)	37.5 (13.65–110.1)

SUV: Standardized Uptake Value; MTV: Metabolic Tumor Volume; TLG: Tumor Lesion Glycolysis.

**Table 3 cancers-14-00637-t003:** Association between the four PET/CT prognostic features extracted from the correlation matrix with clinical and histological parameters. The *p*-value was calculated from a linear regression model with the Wald test. A value of *p* < 0.05 was considered statistically significant.

Variables	Homogeneity *p*-Value	MTV *p*-Value	Entropy *p*-Value	SRE *p*-Value
T3–T4 vs. T1–T2	0.614	<0.001	<0.001	0.545
cN+ vs. cN0	0.403	<0.001	0.027	0.713
N+_PET_ vs. N0_PET_	0.037	<0.001	0.001	0.164
Inflammatory tumor	0.117	<0.001	<0.001	0.511
mSBR 3 vs. 2	0.004	0.321	0.020	0.044
Unifocal vs. multifocal	0.522	0.094	0.661	0.666

Green = negative statistically significant association. Red = positive statistically significant association.

**Table 4 cancers-14-00637-t004:** Univariate analysis using a Cox model of clinical, histological and PET/CT parameters with DFS and OS for the 111 TNBC patients.

Variables	DFS		OS	
	HR (95% CI)	*p*	HR (95% CI)	*p*
**Tumor characteristics**				
Inflammatory: yes	**2.79 (1.14; 6.81)**	**0.025**	**4.53 (1.61; 12.78)**	**0.004**
T3–T4 vs. T1–T2	1.56 (0.77; 3.17)	0.220	**2.78 (1.16; 6.69)**	**0.022**
cN+ vs. cN0	**3.3 (1.62; 6.71)**	**0.001**	**2.58 (1.06; 6.32)**	**0.038**
mSBR 3 vs. 2	1.47 (0.35; 6.14)	0.599	1.85 (0.25; 13.84)	0.549
Associated in situ carcinoma: yes	1.51 (0.64; 3.53)	0.343	2.27 (0.86; 6.02)	0.099
Ki67 ≥ 20% vs. Ki 67 < 20%	0.63 (0.19; 2.1)	0.452	0.69 (0.16; 3.01)	0.624
Unifocal vs. multifocal tumor	0.51 (0.25; 1.04)	0.066	**0.21 (0.08; 0.5)**	**<0.001**
**Therapy**				
NAC vs. adjuvant CT	1.8 (0.69; 4.69)	0.226	2.78 (0.64; 11.98)	0.171
Without anthracyclines vs. anthracyclines	**16.38 (4.6; 58.37)**	**<0.001**	**36.36 (8.28; 159.64)**	**<0.001**
Platin salts vs. anthracyclines	**3.76 (1.73; 8.16)**	**0.001**	**8.2 (3.07; 21.91)**	**<0.001**
Radical surgery/conservative surgery	1.56 (0.79; 3.09)	0.202	**4.04 (1.61; 10.14)**	**0.003**
pCR vs. absence of pCR	**0.17 (0.04; 0.73)**	**0.017**	0.33 (0.07; 1.45)	0.141

NAC: neoadjuvant chemotherapy; CT: chemotherapy; pCR: pathological complete response.

**Table 5 cancers-14-00637-t005:** Univariate analysis using a Cox model of PET/CT features of the primary breast tumor with DFS and OS for the 111 TNBC patients.

PET/CT Parameters	DFS		OS	
	HR (95% CI)	*p*	HR (95% CI)	*p*
N+_PET_ vs. N0_PET_	**2.91 (1.42; 5.97)**	**0.004**	**3.85 (1.4; 10.59)**	**0.009**
SUV_max_	1.03 (0.99; 1.07)	0.131	1.04 (0.99; 1.09)	0.153
SUV_mean_	1.07 (0.92; 1.24)	0.361	1.11 (0.92; 1.34)	0.267
SUV_peak_	1.03 (0.98; 1.09)	0.260	1.04 (0.97; 1.12)	0.232
**MTV**	**1.39 (1.1; 1.76)**	**0.006**	**1.7 (1.24; 2.33)**	**0.001**
**TLG**	**1.31 (1.07; 1.6)**	**0.008**	**1.53 (1.17; 2.01)**	**0.002**
**Entropy**	**2.08 (1.1; 3.96)**	**0.025**	**2.46 (1.02; 5.97)**	**0.046**
Homogeneity	0.19 (0; 10.58)	0.421	0.13 (0; 30.74)	0.469
SRE	7.49 (0; 88,438,904.54)	0.809	6.79 (0; 24,884,680,339.1)	0.865
LRE	1.38 (0.03; 57.66)	0.864	2.33 (0.02; 277.48)	0.729
LGZE	0.47 (0.18; 1.21)	0.118	0.34 (0.09; 1.25)	0.105
HGZE	1.48 (0.96; 2.3)	0.078	1.61 (0.91; 2.86)	0.101

**Table 6 cancers-14-00637-t006:** Multivariate analysis of PET/CT features and tumor characteristics for the 111 TNBC patients with DFS and OS. The variables included in the “full” multivariate model are MTV, TLG, entropy, SUV_max_, N0_PET_/N+_PET_, inflammatory tumor (inflam) and T3–T4/T1–T2 stage.

Parameters	DFS		OS	
	*p*-Value	HR (95% CI)	*p*-Value	HR (95% CI)
MTV	0.125		**0.014**	**1.52 (1.09; 2.12)**
TLG	0.158		-	
Entropy	0.192		-	
SUV_max_	-		-	
N+_PET_ vs. N0_PET_	**0.036**	**2.36 (1.06; 5.24)**	0.074	
Inflam	-		-	
T3–T4 vs. T1–T2	-		-	

## Data Availability

Data are contained within the article.

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
