# Peer review of "Prognostic Value of Metabolic, Volumetric and Textural Parameters of Baseline [18F]FDG PET/CT in Early Triple-Negative Breast Cancer"

_cancers, 2022, doi:10.3390/cancers14030637_

Round 1

Reviewer 1 Report

Thank you for the opportunity to revise your manuscript. Breast cancer is one of the most common cancer type among women, and appropriate diagnosis might be beneficial for the patients. However, this manuscript requires additional clarifications, verification and comparison. This article currently lacks rigorous description. The following issues must be addressed:

Major comments:
1. Please provide changes of the nomenclature in whole manuscript according to "Consensus nomenclature rules for radiopharmaceutical chemistry — setting the record straight"
2. Simple summary should not sound like the aim. This should be short summary of the major findings.
3. In the abstract, in results section: please add values for ‘High TLG, MTV and entropy”.

  1. Authors indicate that they evaluate ‘metabolic parameters”, however from all assesses PET-derived parameters, only SUV is known as a ‘metabolic parameter”. Other two: TLG and MTV are volumetric parameters.
  2. Why patients with SUV values were excluded from the analysis? Does they have TNBC?
  3. Material and Methods section, paragraph “18F-FDG PET/CT examination”: please add information about slice thickness for low dose CT as well as the time for bed scan in PET studies.
  4. Material and Methods: The phantom study – in the TEP/CT 3 there is an extra two dots with high activity, which are not visible on TEP/CT 1 and TEP/CT 2. Could authors explain what might be the cause/reason for this extra two dots on TEP/CT 3?
  5. Material and Methods, section “18F-FDG PET/CT imaging analysis”: please add the time of experience of the nuclear medicine physician.

9 Material and Methods, section “18F-FDG PET/CT imaging analysis”: Why authors decided to use the threshold of 2.5 of SUV? This is not commonly used in PET studies. According to the EANM the 41% SUV is preferable.

  1. Material and Methods, section “18F-FDG PET/CT imaging analysis”: which entropy Authors used? Shannon?
  2. Material and Methods, section “18F-FDG PET/CT imaging analysis”: The multifocal breast tumor – this should be assessed as a separate group, because of the worse prognosis. Moreover, please indicate the number for patients with multifocal and bilateral breast cancer.
  3. Material and Methods, section “Statistical analysis”, Paragraph 3 and 4 – Why Authors in this section put the Aim of the study? These two statements should not be included in the “statistical analysis” section.
  4. Material and Methods, section “statistical analysis”: from where Authors had information/confirmation of patients death? Does it was taken from the National Cancer Registry?
  5. Material and Methods, last paragraph: there are some works, where authors indicate that in terms of radiomics the p value less than 0.005 should be considered significant. Please explain why you decided to use only p 0.05?
  6. Results, paragraph 3.1 – it will be more readable for the reader if authors use STARD Flow diagram indicating number of patients, inclusion and exclusion criteria.
  7. Results, paragraph 3.2: why author checked only 1 metabolic parameter? Which one?
  8. Results, paragraph 3.3, Table 1: please add the number of patients who received chemo-radiotherapy (CRT) and the range for the age of the patients.
  9. Results section: Does N+PET LN were confirmed either by dissection or biopsy? Or were only assessed on PET studies?
  10. Results, paragraph 3.4: inflammatory carcinoma as well as MTV are well known prognosticators for cancer patients. Moreover, in case of inflammatory carcinoma – the MTV values might be higher because of the major limitation of [18F]FDG – it accumulates in inflammatory cells, and it is hard to distinguish between inflammation and carcinoma. Does Authors in this case performed a DTP scan? Which is helpful in discrimination between these two types?
  11. In all Tables there is no explanation for the abbreviation used in tables in the whole manuscript.
  12. Results, section 3.5: In the manuscript authors present two “tables 5” – they should be connected into one table, which will be more scientific.
  13. Results section: why authors decided to use in multivariate analysis only PET parameters and inflammation, if they indicate in univariate analysis that there are also other significant parameters, like therapy type?

Minor comments:

  1. References should be placed in the square brackets.
  2. SUVmax and SUVmean values are usually written to one decimal place
  3. SUV should be written as “standardized uptake value”, not “standard uptake value”
  4. Once Authors write “Jezequel et al.” once “Kim”, “Ohara” – please make the references homogenous.
  5. Please check carefully whole manuscript, because some abbreviations like PFS are not explained in the text.

Author Response

Kind regards

Reviewer 2 Report

This manuscript is an interesting study about the association between metabolic parameters and textural features at FDG PET/CT performed in triple-negative breast cancer (BC) patients at staging.

The paper is of interest to readers, here below my suggestions to improve the work.

The Introduction section should be more focused on radiomics analysis and its opportunities in clinical evaluation rather than on the characteristics of triple-negative BC. 

Did you try to use a threshold of 41% of SUVmax instead of a cut-off of 2.5?

In the Discussion section you mention study on radiomics analysis applied to other types of cancer; in my opinion, instead, should be interesting discussing the other papers about the same topic published so far (e.g. the most recent wih PMID: 34374796, 34328530, 34208197 and many more).

Author Response

Kind regards

Reviewer 3 Report

The goal of this study is very interesting, important, and is well in the field of nuclear medicine.  Especially when,  it is estimated that triple-negative breast cancer (TNBC) accounts for approximately 15-20% of all cases of this disease. The disease is characterized by an unfavorable clinical course and a poor prognosis. All these factors show the importance of early diagnosis and initiation of treatment if we already know that the FDA approved this year pembrolizumab for high-risk, early-stage, TNBC in combination with chemotherapy as neoadjuvant treatment before and after surgery.

In general, the manuscript is well designed,  the studies have been properly conducted and the obtained results are interesting. Overall this paper could be accepted after the minor revision noted below.

  • General comment, please follow the EANM GUIDANCE and please change 18F-FDG  to
    2-[18F]FDG or [18F]FDG; please see H.H. Coenen et al. Nuclear Medicine and Biology 55 (2017) v–xi; https://www.eanm.org/content-eanm/uploads/2019/12/EANM_GUIDANCE-_TRACER_NOMENCLATURE.pdf
  • The literature is not properly cited e.g. In the text, reference numbers should be placed in square brackets [ ] instead of round brackets ( ). The Authors do not follow the Cancers references standard:  Author 1, A.B.; Author 2, C.D. Title of the article. Abbreviated Journal Name YearVolume, page range.
  • - the quality of Figures 7 to 9 should be improved

Author Response

Kind regards

Round 2

Reviewer 1 Report

Thank you for the opportunity to review a revised manuscript “Prognostic value of Metabolic, Volumetric and Textural Parameters of Baseline [18F]FDG PET/CT in Early Triple-Negative Breast Cancer.’ Authors provided an appropriate changes, however, still some minor changes are needed before accepting:

1) “Response 9: Thank you for this comment. Even if EANM recommend, when possible, the use of a threshold of 41%SUV, we realized that this method was not clearly adapted for highly metabolic lesions. At the beginning of this study, we used this method and some lesions were not well segmented. In the literature, 2 studies of Kim et al. and Yue et al. have evaluated TNBC using SUV2.5 threshold, which was more satisfying (references added in “Materials and Methods section – [18F]FDG imaging analysis” and we have made comments in discussion part).-> If Authors provided such an analysis please add this data where you compare these 2 methods (SUV 2.5 and Th41%) in the manuscript. This will raise the scientific level of the article.

2) “Response 13: We have information of patient’s death in medical file with the death certificate. Patients were followed by oncologists in our Institute every 6 months during 5 years.’ -> Please add this information in the appropriate section of the manuscript.

3) “Response 14: As suggested by Lambin et al. [ref39], radiomics works would address the association between high-throughput extraction of large amounts (>200) of quantitative features of medical images and clinical assessments. Therefore, like in genomics studies, the number of studied parameters may often exceed the number of observations (p>>n), which requires specific statistical solutions, including strong reduction of p-value threshold for example. In this study, we have focused on very fewer parameters, and since n>>p, the conventional statistical methods may be applied. Nevertheless, rather than acting a statistical conclusion, we aim to explore the possible prognostic value of some metabolic parameters and textural features in specific TNBC population at the initial examination. Do we have to insert this explanation in the discussion?” -> Yes, please add this information in the discussion section.

Author Response

Thank you for these comments.
